# People Cheat on Task Performance When They Feel Bored: The Mediating Role of State Self-Efficacy

**DOI:** 10.3390/bs12100380

**Published:** 2022-10-03

**Authors:** Chun Feng, Chuanjun Liu, Min Zhong

**Affiliations:** 1Department of Applied Psychology, Faculty of Law, Southwest University of Science and Technology, Mianyang 621010, China; 2Department of Sociology and Psychology, and Institute of Psychology, School of Public Administration, Sichuan University, Chengdu 610065, China; 3Department of Medical Technology, Sichuan Nursing Vocational College, Chengdu 610000, China

**Keywords:** state boredom, cheating behaviors, state self-efficacy

## Abstract

It is unclear whether the state of boredom is related to morality. The present study investigated how state boredom influenced cheating behaviors on task performance. In Study 1 (*N* = 104), participants were induced to feel bored, and then reported whether they had finished an anagram task (two sentences in the task were unsolvable). The results found that people with higher boredom showed more cheating behaviors than those with lower boredom on task performance. In Study 2 (*N* = 139), participants completed the Multidimensional State Boredom Scale, and then completed the same anagram task as in Study 1, as well as a state self-efficacy scale. The results revealed that state self-efficacy mediated the effect of state boredom on cheating behaviors on task performance. In other words, a higher level of state boredom leads to a lower level of state self-efficacy, and the lower state self-efficacy then results in more cheating behaviors. The present study provides empirical evidence that state boredom has its moral function through state self-efficacy.

## 1. Introduction

The state of boredom is generally regarded as a concrete and short-lived emotion [1,2], which is “*the actual experience of boredom in a given moment*” (p. 70) [3]. Retana proposed that boredom is a moral emotion [4]. Similarly, Van Tilburg and Igou examined the positive correlation between state boredom and prosocial intentions, revealing the moral implications for the state of boredom [5]. However, Van Tilburg and Igou also indicated that boredom is perceived as an unrelated emotion to morals compared to other negative emotions (e.g., sadness, anger, disgust, etc.) [6].

Notably, the findings from Van Tilburg and Igou remain a limitation. They asked participants to read the description, “Some emotions are associated with morality (i.e., these emotions relate to the question of what makes a good or bad person) whereas other emotions are not associated with morality” (p. 12) [6]. Participants then evaluated whether the emotions (including boredom) were related to morality. Indeed, the method of measuring whether boredom is related to morality was biased, because the method of measurement was subjective and did not measure the causal relationship between boredom and moral behaviors. Thus, it was not sufficient to conclusively support the relationship between state boredom and morality.

Given the inconsistent findings of previous studies and biased measurement, it is unclear whether the state of boredom is related to morality. Importantly, the relationship between state boredom and cheating behaviors so far has received little attention. Taken together, the present study mainly concerns cheating behaviors, extending previous studies on the correlation between state boredom and morality by providing novel empirical evidence.

The current work has two primary goals. The first goal is to measure whether state boredom can predict cheating behaviors. The second goal is to find the underlying mechanism, namely the mediating role of state self-efficacy between state boredom and cheating behaviors. The current work would indicate the theoretical implications of linking state boredom to cheating behaviors. We begin with a literature review on state boredom and cheating behaviors.

### 1.1. State Boredom

Previous researchers have proposed various definitions of state boredom. Pekrun et al. indicated the different aspects of state boredom: “*specific affective components (unpleasant, aversive feelings), cognitive components (altered perceptions of time), physiological components (reduced arousal), expressive components (facial, vocal, and postural expression), and motivational components (motivation to change the activity or to leave the situation)*” (p. 532) [2]. In the present study, we regarded boredom as an activity-related emotion, which is most relevant for one’s task performance [7,8,9].

### 1.2. State Boredom vs. Trait Boredom

State boredom is related to trait boredom but is a distinct construct from trait boredom [10,11,12]. Specifically, trait boredom is an internal psychological characteristic, which focuses on one’s tendency to become bored. In contrast, state boredom is “*the actual experience of boredom in a given moment*” (p. 70) [3], which is more influenced by external situational factors [3]. The issue of boredom has been discussed by Neu in terms of endogenous boredom (boredom that stems from within) and reactive boredom (boredom that stems from the environment) [11]. Similarly, Todman also has distinguished boredom as situation-independent boredom and situation-dependent boredom [12]. Accordingly, researchers developed the Boredom Proneness Scale (BPS) to measure trait boredom [13] and the Multidimensional State Boredom Scale (MSBS) [3] to measure state boredom [3].

We are interested in state boredom, because boredom sometimes occurs in situations of repetition, meaningless tasks, or too little challenge given one’s skills [14,15,16,17,18], which is an unpleasant experience and always has negative consequences as a result. For instance, cheating behavior is one of the moral behaviors that is a widespread phenomenon [19] and has been found to be associated with trait boredom (e.g., organizational misbehaviors). However, the correlation between state boredom and cheating remains unclear. To the best of our knowledge, no prior studies have focused on the correlation. Thus, in the present study, we focused on the relationship between state (rather than trait) boredom and cheating behaviors.

### 1.3. Boredom and Cheating Behaviors

Some researchers have measured the correlation between trait boredom and moral disengagement at the workplace. For instance, people who are highly bored on the job have significantly higher scores on an objective measure of nonappearance in a job [20]. Similarly, highly bored employees are more likely to engage in organizational misbehaviors (e.g., taking long breaks at work, accepting bribes, misusing company resources, threatening colleagues, etc.) [21,22]. Given that state boredom and trait boredom are related constructs [10], we predicted that people with higher levels of state boredom would be more likely to cheat than those with lower levels of state boredom (Hypothesis 1).

### 1.4. State Self-Efficacy vs. Trait Self-Efficacy

Bandura identified state self-efficacy as an individual’s belief about their capability to accomplish specific performance goals based on an individual’s expectations and convictions about what he or she can accomplish under specified circumstances [23,24,25]. Thus, research on state self-efficacy emphasizes the contextual role in assessing efficacy [26]. Meanwhile, trait self-efficacy refers to the way in which one perceives himself or herself in particular domains of activity, which is closer to the general self-concept [26]. For instance, when it comes to academic domains, trait self-efficacy refers to how competently a person perceives themselves in academic settings, which is fairly stable [26]. In contrast, state self-efficacy refers to a person’s belief that he or she is capable of successfully completing academic tasks at a designated level [27], which is relatively malleable based on the nature of the task.

As indicated above, state self-efficacy (vs. trait self-efficacy) is more determined by a specific situation or under a certain task. We were mainly concerned with whether state boredom would decrease state self-efficacy when participants completed an unsolvable task in the present study. Thus, we predicted that the underlying mechanism may be state (rather than trait) self-efficacy.

### 1.5. Linking State Boredom and Cheating Behaviors: The Mediating Effect of State Self-Efficacy

Boredom is regarded as a “gadfly sting” that causes people to realize that they are incapable of engaging successfully in worthwhile pursuits. Previous research has indicated that boredom is negatively correlated with self-efficacy. For instance, theorists proposed that higher levels of boredom would allow people to process information less efficiently, gain less competence, and succeed less at academic tasks [8,28]. Similarly, research yielded empirical evidence for negative correlations between academic-related boredom and academic self-efficacy [29,30,31,32]. Specifically, repetitive activities might lead to feeling bored. In turn, state boredom decreases the level of activation on the task according to the measurement of heart rate or skin conductance [1], i.e., decreasing people’s concentration on the task [7], a sense of control (e.g., self-efficacy) over the task [31], or a fear of failure [33], etc. Moreover, boredom was negatively associated with students’ self-perceived competence and their performance [34]. In conclusion, state boredom may be negatively correlated with state self-efficacy.

In addition, some evidence suggests that the likelihood of cheating is lower among students with confidence in their academic abilities [35]. Murdock et al. reported that the perception of academic self-efficacy is negatively related to cheating in school [36]. The correlation between self-efficacy and cheating has also been found in college samples [37]. It is because students with low self-efficacy “may doubt their ability to bring about a desired result, which may lead to reliance on other strategies for success” (p. 109) [36], i.e., cheating behaviors [37].

Although previous studies found correlations between boredom, self-efficacy, and cheating behaviors, previous research did not distinguish between state boredom and trait boredom, nor between state self-efficacy and trait self-efficacy. It is unclear whether and how state boredom, state self-efficacy, and cheating behaviors are correlated with each other. We predicted that state self-efficacy might play a mediating role between state boredom and cheating behaviors (Hypothesis 2).

### 1.6. Present Study

In the current work, we primarily aimed to measure the correlations between state boredom, state self-efficacy, and cheating behaviors on task performance. In Study 1, participants’ state boredom was manipulated before they took a cheating behavior test. In a randomized experiment, we tested whether the cheating behavior differed between high-state boredom and low-state boredom. In Study 2, we used the MSBS [3] to measure state boredom and included measures of state self-efficacy as a mediating variable. We expected state boredom to be related to cheating behaviors on task performance. We also expected that the effect of state boredom on cheating behaviors would be mediated by state self-efficacy.

## 2. Study 1 Does Cheating Behavior Differ between High-State Boredom and Low-State Boredom?

Study 1 examined whether cheating behavior differed between high-state boredom and low-state boredom on task performance. In this study, participants were induced to feel bored, and then reported whether they had finished an anagram task (two sentences in the task were unsolvable).

### 2.1. Method

#### 2.1.1. Participants

G*Power software version 3.1 [38] was used to calculate the required sample size. We estimated, based on our theorizing, that the difference in cheating behaviors regarding a high level of boredom would be moderate (Cohen’s *d* = 0.50). Power was set to 0.80, as recommended by Cohen [39]. In addition, the alpha was set at 0.05. The analysis revealed that we needed 102 participants in total. We recruited 104 Chinese students (49 males and 55 females) by advertisement from a university in Sichuan province, China. Ages ranged from 18 to 23 (*M* = 19.87, *SD* = 1.28, 3 participants did not report their age). Participants were randomly assigned to the high-boredom (*N* = 47) or low-boredom condition (*N* = 57). Participants received a gift for their participation.

#### 2.1.2. Measurement and Procedure

The study was approved by an ethical panel at the Psychology Department at Southwest University of Science and Technology in Sichuan, China. Participants signed consent forms.

Manipulated boredom experience. At the beginning of the study, participants were randomly assigned to the high-boredom or low-boredom condition according to the last number of their student ID. Even numbers were assigned to the high-boredom condition and odd numbers were assigned to the low-boredom condition. Participants in the high-boredom condition completed the highly boring task of copying ten references, whereas those in the low-boredom condition completed the less boring task of copying two references. The boredom manipulation was adapted from Van Tilburg and Igou [18]. Then, they reported the extent to which they felt bored (1 = “not at all”, 7 = “very much”) after the reference copying task.

Measures of cheating behaviors. Participants were asked to complete an anagram task, which we adapted from Kilduff et al. [40] and Pierce et al. [41]. We instructed participants to attempt to solve four sentences, in which we broke down each word into a syntactically illogical sequence (i.e., “on the playground, Zhang Liang, after class, played”, “some, white things, he, on the ground”, “mathematician, self-taught, is a, famous, Hualuo Geng”, and “the heat, I am happy, in my hand, to pick”), in three minutes and indicated whether they completed each sentence on a yes/no response scale. Participants were told that they would receive more gifts at the end of the experiment if they solved more sentences correctly. Indeed, the first and third sentences could be solved logically, whereas the second and fourth sentences had no solution. Thus, those who reported to have solved neither the second nor the fourth sentence were coded as 0, those reported to have solved either the second or fourth sentence were coded as 1, and those who reported to have solved both the second and fourth sentences were coded as 2, with a higher score indicating more cheating behaviors. At the end of the experiment, participants were told that the amounts of gifts that they received were not related to the anagram task.

### 2.2. Results

Boredom manipulation check. We first coded the boredom condition as low = 0 and high = 1. Then, we entered the boredom feeling scores as the dependent variable and the boredom condition as the independent variable in an independent-sample T test. The results indicated that participants in the high-boredom condition felt more bored (*M* = 4.96, *SD* = 2.04) than those in the low-boredom condition (*M* = 3.49, *SD* = 1.76), *t* (100) = 3.81, *p* < 0.001, 95% CI = [0.72, 2.21], Cohen *d* = 0.77, as shown in Table 1.

Cheating behaviors. We entered the cheating behaviors as a dependent variable and the boredom condition as an independent variable into an independent-sample *t* test. The results indicated that participants in the high-boredom condition showed more cheating behaviors (*M* = 1.47, *SD* = 0.75) than those in the low-boredom condition (*M* = 1.05, *SD* = 0.74), *t* (102) = 2.83, *p* < 0.01, 95% CI = [0.12, 0.71], Cohen’s *d* = 0.56, as shown in Table 1.

Correlation analysis. Age and gender were not significantly correlated with cheating behaviors in the low- (*r* = −0.18, *p* = 0.20; *r* = −0.06, *p* = 0.68) and high-boredom conditions (*r* = −0.07, *p* = 0.66; *r* = 0.12, *p* = 0.45). See Table 2.

In addition, we obtained the correlations between the manipulation check and cheating behaviors within each condition and across the two conditions. Results showed that the correlation between the manipulation check and cheating behaviors was not significant in the high-boredom condition (*r* = 0.10, *p* = 0.50), in the low-boredom condition (*r* = −0.16, *p* = 0.25), and across the two conditions (*r* = 0.07, *p* = 0.47). A possible explanation is that when participants answered the manipulation check questions, their mental processes may have been affected in ways specific to that particular setting [42]. However, this affection is particularly difficult to identify [43]. Hauser et al. suggested that when manipulation checks are used, mediation analyses may not be able to eliminate confounding variables, since variables unmeasured may still influence the outcome [42]. Thus, the effect of the manipulation check on the outcome would be biased. This could be the reason that we did not find a correlation between the manipulation check and cheating behaviors.

### 2.3. Brief Discussion

Consistent with Hypothesis 1, Study 1 revealed that participants in the high-boredom condition showed more cheating behaviors than those in the low-boredom condition on task performance. In order to replicate the findings in Study 1, we tried to use different measurements to examine the correlation between state boredom and cheating behaviors. Thus, in Study 2, we applied the MSBS [3] to measure the current boredom state of participants. In addition, we further explored the underlying mechanism for the effect of state boredom on cheating behaviors, namely state self-efficacy.

## 3. Study 2 Is the Effect of State Boredom on Cheating Behaviors Mediated by State Self-Efficacy?

In Study 2, we examined whether the effect of state boredom on cheating behaviors would be mediated by self-efficacy on task performance. Participants in Study 2 completed the MSBS [3], and then completed the same anagram task as in Study 1, as well as a state self-efficacy scale.

### 3.1. Method

#### 3.1.1. Participants

We estimated the sample size based on previous research [33,44]; the effect of boredom on self-efficacy was of a large size (*β* = 0.50) and the effect of self-efficacy on cheating behaviors was approximately halfway between a small (*β* = 0.14) and medium (*β* = 0.39) effect (*β* = 0.29). In order to achieve 0.8 power to find the mediation effect at α = 0.05, we required 121 participants [45]. In the end, we recruited 139 participants (39 males and 100 females) from a university in Sichuan province, China. Ages ranged from 18 to 22 (*M* = 19.54, *SD* = 0.89, 3 participants did not report their age). Participants received a gift for their participation.

#### 3.1.2. Procedure

The study was approved by the same ethical panel as in Study 1. Participants signed consent forms.

Boredom. Participants completed the MSBS, which was developed by Fahlman et al. [3]. The MSBS is a 30-item questionnaire on a 7-point Likert scale ranging from 1 (= strongly disagree) to 7 (= strongly agree). An example statement is ‘*I am stuck in a situation that I feel is irrelevant*’. The MSBS had acceptable reliability (*α* = 0.95) in the present study.

Cheating behaviors. The measurement was identical to that in Study 1.

State self-efficacy. Because we aimed to measure state self-efficacy according to the task of cheating behaviors in our study, we adapted the general self-efficacy scale [46] by revising the instruction of self-efficacy according to the task of cheating behaviors. Specifically, we asked participants to indicate their current belief in their capability to accomplish the anagram task above. The scale contains a 10-item scale. An example statement is ‘*I am confident that I could deal efficiently with unexpected events*’ (1 = Not at all true, 4 = Exactly true). The reliability in the present research was high (*α* = 0.91).

### 3.2. Results

Common method bias test. A common method bias test was conducted using the Harman single-factor test [47]. The results showed that the KMO value was 0.89 (*p* < 0.001), indicating that the scales were suitable for factor analysis. There were eight factors with eigenvalues greater than 1, and the first factor explained a total variance of 34.10%, which did not reach the critical criterion of 40%. Therefore, the influence of common method bias was not considered to be great in this study.

Correlation analysis. As expected, state boredom was negatively related to self-efficacy (*r* = −0.43, *p* < 0.001) but positively correlated with cheating behaviors (*r* = 0.20, *p* = 0.02). Self-efficacy was negatively related to cheating behaviors (*r* = −0.30, *p* < 0.001). Regarding demographics, with the exception of age, which was correlated with state boredom (*r* = −0.17, *p* = 0.05), age and gender were not significantly related to any other variables. See Table 3.

A mediation analysis using 5000 bootstrapping samples [48] was conducted to examine the proposed mediation model, with multidimensional state boredom as the predictor, self-efficacy as the mediator, and cheating behaviors as the outcome. See Figure 1. The results showed that the effects of state boredom on self-efficacy (*B* = −0.37, *SE* = 0.07, *t* = −5.50, *p* < 0.001, 95% CI = [−0.50, −0.24]) and cheating behaviors (*B* = 0.16, *SE* = 0.07, *t* = 2.33, *p* = 0.02, 95% CI = [0.30, 0.02]) were significant. The effect of state self-efficacy on cheating behaviors was significant, *B* = −0.24, *SE* = 0.09, *t* = −2.86, *p* < 0.01, 95% CI = [−0.41, −0.08]. The indirect effect of state boredom on cheating behaviors, via state self-efficacy, was significant, *B* = 0.09, *SE* = 0.03, 95% CI = [0.03, 0.17]. The direct effect of state boredom on cheating behaviors was not significant, *B* = 0.07, *SE* = 0.07, *t* = 0.94, *p* = 0.35, 95% CI = [−0.08, 0.22]. The total effect of state boredom on cheating behaviors was negative, *B* = 0.16, *SE* = 0.07, *t* = 2.33, *p* = 0.02, 95% CI = [0.02, 0.30].

### 3.3. Brief Discussion

Consistent with Hypothesis 2, Study 2 revealed that state boredom was indirectly associated with cheating behaviors on task performance through reduced state self-efficacy.

## 4. General Discussion

In the present work, we conducted two studies to examine the correlation between state boredom and cheating behavior on task performance and their underlying mechanism, namely state self-efficacy. The studies firstly provided evidence that people with higher state boredom showed more cheating behaviors than those with lower state boredom on task performance (Study 1). The findings were consistent with the existing literature on the correlation between trait boredom and organizational misbehaviors at work [20,21,22]. The present results also found that state boredom was indirectly associated with cheating behaviors on task performance through reduced state self-efficacy (Study 2). The findings were also consistent with the existing literature. For instance, academic-related boredom was negatively correlated with academic self-efficacy [30,37]. In addition, self-efficacy is negatively related to cheating [35,36].

Our correlation analyses in Study 2 showed that gender was not significantly correlated with state self-efficacy, which is inconsistent with some previous studies, in which males had significantly higher general self-efficacy than females [49,50,51] because males are sometimes overconfident of their abilities and performance when self-recognizing, whereas females show the opposite [52]. However, our findings are consistent with other previous studies. According to Hackett and Campbell, gender-neutral tasks did not reveal any differences between men and women, and the possible reason is that the nature of the task may influence the self-efficacy expectations [53]. For instance, there is a greater likelihood of gender differences in self-efficacy where there is more gender-role stereotyping in tasks (such as math examinations) [54,55,56]. Thus, we can presume that the inconsistency between our findings and previous findings may also be related to the nature of the task, because our cheating task was not relevant to traditional gender-role stereotyping. Consequently, we did not find a significant correlation between gender and self-efficacy.

Our findings have important theoretical implications. This study first provides empirical evidence for the relationship between state boredom and cheating behaviors, which represents a limitation of previous research [5]. Our investigation extends previous studies on the correlation between state boredom and morality by providing novel empirical evidence.

Second, although previous studies found that trait boredom was positively associated with organizational misbehaviors, the present study extends previous research and highlights the moral function of state boredom by studying the relationship between state boredom and cheating behaviors on task performance. Furthermore, Elpidorou proposed that boredom becomes a moral issue when it is regarded as a character trait [57]. The present study provided empirical evidence that state boredom also has its moral function, even if it is not a character trait. Thus, our findings help to advance research on boredom.

Third, previous studies found correlations between boredom, self-efficacy, and cheating behaviors. However, previous research did not distinguish between state and trait boredom, nor between state self-efficacy and trait self-efficacy. It was unclear whether and how state boredom, state self-efficacy, and cheating behaviors were correlated with each other. In the present study, we fill this gap by making a distinction between state boredom and trait boredom, and between state self-efficacy and trait self-efficacy. We found that state self-efficacy plays a mediating role between state boredom and cheating behaviors.

Our findings also carry important practical implications. Individuals may experience boredom for many reasons in everyday life, because life is not always filled with happiness, interest, and excitement; rather, it is often routine, dull or boring. In this situation, when individuals with a high level of boredom are requested to perform a task, their state boredom increases their cheating behavior through reduced state self-efficacy. Thus, our findings could offer an implication for schools or enterprises, i.e., it may be necessary to provide an engaging environment or to inspire individuals to perform tasks that they are interested in during daily work or practice, which would prevent them from acting immorally.

The present study had a few limitations. First, the participants were all undergraduate students, well-educated, and mostly under the age of 22. The participants might have had a routine and simple life at university, so they may have possessed a different perception of boredom from other populations. It would be worthwhile to examine whether the correlation between boredom and cheating behaviors exists among the general population.

Second, although the anagram task, which we adapted from Kilduff et al. [40] and Pierce et al. [41], is commonly used to measure immoral behaviors, the measurement could be confounded by participants’ IQ, interests, attention, etc. Further studies could exclude these confounding variables in order to verify the present findings.

Third, the anagram task in the present study was somewhat easy. Indeed, task difficulty is negatively correlated with self-efficacy [58]. Thus, if we increase the difficulty of the task, the correlation between boredom and cheating behaviors might become greater. This is because when a task is more difficult, self-efficacy would be lower, which in turn may lead to an increase in cheating behaviors. Future research could manipulate the difficulty levels of tasks to replicate the present findings.

Fourth, in Study 1, we manipulated the high- vs. low-boredom conditions by asking participants to copy different numbers of (ten vs. two) references. Although the manipulation in Study 1 was successful and we replicated the correlation between state boredom and cheating behaviors in Study 2, participants were required to spend more time and use more cognitive resources to finish the task in the high-boredom condition relative to those in the low-boredom condition. The higher cognitive load of the high- vs. low-boredom condition might be a potential factor driving the result. In addition, when participants copied ten (vs. two) references, they may have experienced higher ego depletion and had less control in regulating their behaviors, which may also have led to cheating behaviors. Previous studies found that ego depletion was positively associated with unethical behavior [59]. Thus, it would be worthwhile to measure ego depletion and its role in a future study to verify the present study.

However, the manipulation of boredom and ego depletion is different; for instance, the task of manipulating ego depletion (e.g., the letter “e” task) is challenging and requires attention to detail. In contrast, the task of copying ten (vs. two) references, which is a classical boredom task, is easier, more repetitive, and less challenging. It has been hypothesized that manipulating ego depletion (letter “e” task) may induce boredom, but less than the task’s directly induced boredom [60], suggesting that there might be a confounding effect of boredom on ego depletion [60,61,62]. Notably, so far, it is not clear whether the manipulation of boredom (ten vs. two copying task) would trigger ego depletion. This is an interesting issue, which could be further measured in the future.

Finally, we primarily aimed to measure the correlation between state boredom and cheating behaviors on task performance, instead of making broad inferences about the correlation by and large. Thus, the present reanalysis mainly illustrates the correlation in the study area of task performance.

## 5. Conclusions

People with higher boredom showed more cheating behaviors than those who experienced lower boredom on task performance. State boredom was indirectly associated with cheating behaviors on task performance through reduced state self-efficacy. The present study provides empirical evidence that state boredom has its moral function through state self-efficacy.

## Figures and Tables

**Figure 1 behavsci-12-00380-f001:**
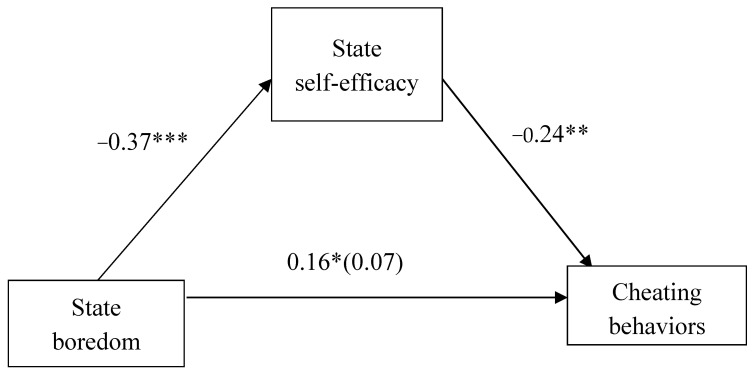
Regression coefficients for the correlation between state boredom and cheating behaviors mediated by state self-efficacy. * *p* < 0.05, ** *p* < 0.01, *** *p* < 0.001.

**Table 1 behavsci-12-00380-t001:** Demographic characteristics of participants.

	High-Boredom Condition *M* (*SD*)	Low-Boredom Condition *M* (*SD*)	Range	*p*	Cohen’s *d*
Boredom manipulation check	4.96 (2.04)	3.49 (1.76)	1–7	<0.001	0.77
Cheating behaviors	1.47 (0.75)	1.05 (0.74)	0–2	<0.01	0.56
Gender	--	--	--	--	--
Age	19.85 (1.33)	19.89 (1.24)	18–23	0.87	0.03

**Table 2 behavsci-12-00380-t002:** Pearson’s correlations of all variables in Study 1 (*N* = 104).

	1	2	3	4
High-boredom condition (*N* = 47)				
1. Boredom manipulation check	1			
2. Cheating behaviors	0.10	1		
3. Gender	0.12	−0.21	1	
4. Age	−0.07	0.12	−0.001	1
Low-boredom condition (*N* = 57)				
1. Boredom manipulation check	1			
2. Cheating behaviors	−0.16	1		
3. Gender	−0.06	−0.02	1	
4. Age	−0.18	−0.05	0.06	1

Notes: Gender was dummy-coded (female = 1, male = 0).

**Table 3 behavsci-12-00380-t003:** Pearson’s correlations of all variables in Study 2 (*N* = 139).

	*M* (*SD*)	Range	1	2	3	4	5
1. State boredom	3.20 (1.01)	1.15–6.40	1				
2. Sate self-efficacy	3.60 (0.87)	1.30–6.00	−0.43 **	1			
3. Cheating behaviors	1.03 (0.82)	0–2	0.20 *	−0.30 **	1		
4. Age	19.54 (0.89)	18–22	−0.17 *	−0.06	0.11	1	
5. Gender	--	--	0.05	−0.02	0.02	−0.07	1

Notes: **p* < 0.05, ** *p* < 0.01. Gender was dummy-coded (female = 1, male = 0).

## Data Availability

The raw datasets of present study can be retrieved from https://osf.io/yjx58/.

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
