# Peer review of "People Cheat on Task Performance When They Feel Bored: The Mediating Role of State Self-Efficacy"

_behavsci, 2022, doi:10.3390/bs12100380_

Round 1

Reviewer 1 Report

I congratulate the authors of this research for the originality of the topic and the quality of their writing. I find their findings relevant to continue studying the possible relationships between boredom and cheating, as well as with unethical behavior in general. 

To enrich the broad critical apparatus they present, I suggest that you consult the following articles derived from Latin American research.

https://www.redalyc.org/articulo.oa?id=16726244006

https://www.redalyc.org/journal/2270/227065157019/html/

https://www.redalyc.org/articulo.oa?id=44248790013

Author Response

Manuscript: behavsci-1726512

Thank you very much for providing us the opportunity to revise the manuscript. We appreciate the comments and suggestions from the reviewers. For your convenience, the review comments and our replies were appended below. The revisions in the manuscript were also highlighted.

reviewer 1

I congratulate the authors of this research for the originality of the topic and the quality of their writing. I find their findings relevant to continue studying the possible relationships between boredom and cheating, as well as with unethical behavior in general. 

To enrich the broad critical apparatus they present, I suggest that you consult the following articles derived from Latin American research.

https://www.redalyc.org/articulo.oa?id=16726244006

https://www.redalyc.org/journal/2270/227065157019/html/

https://www.redalyc.org/articulo.oa?id=44248790013

Response:

Thanks for the information, we reviewed these abstracts and articles, and cited Retana (2011) in the updated manuscript. Please see p.1.

Reviewer 2 Report

The manuscript entitled "People cheat on task performance when they feel bored: The mediating role of state self-efficacy" tested whether state boredom can lead to immoral behaviors, i.e., cheating behavior. The research question was quite interesting. However, I still have some suggestions and concerns, and please find them below.

1.     The manipulation of state boredom. Study 1 revealed a group difference in the cheating behavior between the high boredom and the low boredom condition. The manipulation seemed successful but the key difference between the high vs. low boredom condition was the number of references participants were asked to copy. The workload was quite different between the two conditions. In the high boredom condition, participants need to spend more time and cognitive resources to finish the task relative to those in the low boredom condition. The more cognitive load of the high vs low boredom condition might be the reason driving the result.

2.     Does individual level state boredom (i.e., rating of feeling bored after the manipulation) predict their cheating behaviors? I recommend the authors report the correlations between the two within each condition and across the two conditions.

3.     The measure/manipulation of boredom in Study 2 was a little bit confusing. Did the authors manipulate state boredom (high vs. low)? If so, they need to report the sample size for each condition, the three variables of different situations, and their correlations within and across conditions. If Study 2 was not a between-subject design, the authors may consider running a control condition as Study 1 did. 

4.     The potential alternative should be discussed, such as ego depletion. Because when people copy 10 vs. 2 references, they are also high in ego-depletion and have less control for regulating their behaviors, which leads to cheating. The authors may need to discuss why it is boredom but not other factors that explain cheating.

Author Response

Manuscript: behavsci-1726512

Thank you very much for providing us the opportunity to revise the manuscript. We appreciate the comments and suggestions from the reviewers. For your convenience, the review comments and our replies were appended below. The revisions in the manuscript were also highlighted.

reviewer2

The manuscript entitled "People cheat on task performance when they feel bored: The mediating role of state self-efficacy" tested whether state boredom can lead to immoral behaviors, i.e., cheating behavior. The research question was quite interesting. However, I still have some suggestions and concerns, and please find them below.

  1. The manipulation of state boredom. Study 1 revealed a group difference in the cheating behavior between the high boredom and the low boredom condition. The manipulation seemed successful but the key difference between the high vs. low boredom condition was the number of references participants were asked to copy. The workload was quite different between the two conditions. In the high boredom condition, participants need to spend more time and cognitive resources to finish the task relative to those in the low boredom condition. The more cognitive load of the high vs low boredom condition might be the reason driving the result.

Response:

Thanks for the suggestion! Following your suggestion, we discussed the alternative (e.g., cognitive load) to influences cheating behaviors in the limitations. Please see p.8-9, where we wrote: in Study 1, we manipulated the high vs. low boredom condition by asking participants to copy different number of (ten vs. two) references, we found that participants in the high boredom condition showed more cheating behaviors than those in the low boredom condition did on task performance. Although the manipulation in Study 1 was successful and we replicated the correlation between stated boredom and cheating behaviors in Study 2, participants need to spend more time and cognitive resources to finish the task in the high boredom condition relative to those in the low boredom condition. The more cognitive load of the high vs low boredom condition might be a potential reason driving the result. In addition, when people copy ten (vs. two) references, they may have higher ego-depletion and have less control for regulating their behaviors, which may also lead to cheating behaviors. Previous studies found that ego depletion was positively associated with unethical behavior (Yam et al. 2014). Thus, it would be worthwhile to measure ego depletion and its role in the future study to verify the present study..

  1. Does individual level state boredom (i.e., rating of feeling bored after the manipulation) predict their cheating behaviors? I recommend the authors report the correlations between the two within each condition and across the two conditions.

Response:

Thanks for the suggestion! Following your suggestion, we reported the correlations between the manipulation check and cheating behaviors within each condition and across the two conditions. Please see p.5 for details. We found that the correlation was not significant in high-boredom condition (r = 0.10, p = 0.50), in low-boredom condition (r = -0.16, p = 0.25) and across the two conditions (r = 0.07, p = 0.47). The possibility is that there is a danger in using manipulation check measures, according to researchers (Hauser et al., 2018). When participants answer the manipulation check questions, their mental processes may be affected in ways specific to that particular setting. Seriously, this affection is particularly difficult to identify (Parrot and Hertel, 1999). Hauser et al., (2018) suggested when manipulation checks are used, mediation analyses may not be able to eliminate confounding variables, since variables unmeasured may still influence the outcome. Thus, the effect of manipulation check on outcome would be bias. This would be the reason why we did not find the correlation between the manipulation check and cheating behaviors.

  1. The measure/manipulation of boredom in Study 2 was a little bit confusing. Did the authors manipulate state boredom (high vs. low)? If so, they need to report the sample size for each condition, the three variables of different situations, and their correlations within and across conditions. If Study 2 was not a between-subject design, the authors may consider running a control condition as Study 1 did. 

Response:

Thanks for your suggestion! We did not manipulate state boredom in Study 2, we used the multidimensional state boredom scale (MSBS), which was developed by Fahlman et al. (2013), to measure state boredom. So, it was not able to run a control condition in Study 2.

  1. The potential alternative should be discussed, such as ego depletion. Because when people copy 10 vs. 2 references, they are also high in ego-depletion and have less control for regulating their behaviors, which leads to cheating. The authors may need to discuss why it is boredom but not other factors that explain cheating.

in Study 1, we manipulated the high vs. low boredom condition by asking participants to copy different number of (ten vs. two) references, we found that participants in the high boredom condition showed more cheating behaviors than those in the low boredom condition did on task performance. Although the manipulation in Study 1 was successful and we replicated the correlation between stated boredom and cheating behaviors in Study 2, participants need to spend more time and cognitive resources to finish the task in the high boredom condition relative to those in the low boredom condition. The more cognitive load of the high vs low boredom condition might be a potential reason driving the result. In addition, when people copy ten (vs. two) references, they may have higher ego-depletion and have less control for regulating their behaviors, which may also leads to cheating behaviors.

Response:

Thanks for the suggestion! Following your suggestion, we discussed the alternative (e.g., cognitive load) to influences cheating behaviors in the limitations. Please p.9, where we wrote: Although the manipulation of boredom and ego-depletion is different. For instance, the task of manipulating ego-depletion (e.g., the letter “e” task) is challenging and required attention to detail. In contrast, the copy ten (vs. two) references task, which is a classical boredom task, is more easy, repetitive and unchallengeable. It has been hypothesized that manipulating ego-depletion (letter "e" task) may induce boredom, but less than the task directly induced boredom (Job et al. 2010), suggesting that there might be a confounding effect of boredom on ego-depletion (e.g., Francis, Milyavskaya, Lin, & Inzlicht, 2018; Job, Dweck, & Walton, 2010; Milyavskaya et al., 2019). Notably, so far it is not clear whether the manipulation of boredom (ten vs. Two copying task) would trigger ego-depletion. It is an interesting issue, which could be further measure in the future.

References

Bong, M., & Skaalvik, E. M. (2003). Academic Self-Concept and Self-Efficacy: HowDifferent Are They Really? Educational psychology review, 15(1), 1-40. doi:10.1023/a:1021302408382

Fahlman, S. A., Mercer-Lynn, K. B., Flora, D. B., & Eastwood, J. D. (2011). Development and Validation of the Multidimensional State Boredom Scale. Assessment, 20(1), 68-85. doi:10.1177/1073191111421303

Farmer, R., & Sundberg, N. D. (1986). Boredome Proness - The Development and Correlates of a New Scale. Journal of Personality Assessment, 50(1), 4-17.

Pajares, F., & Schunk, D. (2005). Self-efficacy and self-concept beliefs. New Frontiers for Self-Research, March H. Craven R, McInerney D (eds.). Greenwich, CT: IAP.

Struk, A. A., Carriere, J. S. A., Cheyne, J. A., & Danckert, J. (2015). A Short Boredom Proneness Scale: Development and Psychometric Properties. Assessment, 24(3), 346-359. doi:10.1177/1073191115609996

Reviewer 3 Report

Summary and overall comment

This is an empirical paper consisting of two studies. The first study (N=104) investigated if there was a relation between moral and boredom, that is, if bored people have lower moral standards shown as cheating behavior in an unsolvable anagram task. Results revealed that this was the case, bored people cheated more in this task. The second study (N=139) found that “the state” of self-efficacy mediated “the state” of boredom, a high level of boredom leads to lower self-efficacy, which in turn leads to increased levels of cheating behavior. From this it is concluded state boredom predicts cheating behavior that in turn is mediated by self-efficacy.

The questions are of academic interest but introduction and aims needs a major revision. The intro is very short and there are many concepts that are not properly explained, and several relations and causations are suggested that are not empirically underpinned, e.g., state boredom vs. trait boredom, situation based (state) self-efficacy vs. trait self-efficacy. Moreover, if these affect each other independently and of utmost importance, the danger in claiming causation between factors that correlates. The distinction between possible correlations and causations is not properly described. Therefore, I advocate a major revision of the manuscript in accordance with the detailed descriptions below found in major and minor comments.

Major and minor Comments

1) The distinction between state boredom and trait boredom should be mad clearer. The Boredom Proness Scale (Farmer & Sundberg, 1986) should be mentioned and preferably the psychometric properties of this scale  (Struk, Carriere, Cheyne, & Danckert, 2015). If boredom is a rather constant trait or if it can differ considerably between situations this has to be taken to account when conclusions are drawn as in the present study. The reference for study 2 works also fine for this purpose (Fahlman, Mercer-Lynn, Flora, & Eastwood, 2011)

2) The distinction between Banduras situation (state) determined self-efficacy and the broader trait self-efficacy that is closer to a more general self-concept must be much clearer, see e.g. (Bong & Skaalvik, 2003; Pajares & Schunk, 2005).

3) If you want to link cheating behaviors to one or both above-described concepts there must be room for alternate, third variable, explanations, in particular if you experimentally induce both boredom – through the delivery of boring tasks and cheating behavior through unsolvable tasks. Will these findings be possible to replicate or are they just valid for this particular situation. These clarifications/distinctions should be made clear in the introduction, and alternate explanations should be presented in the discussion.

4) Regarding, the self-efficacy scale neither the scale or the reference is not presented in text or in reference list. Here we are talking about a general trait I guess (not Bandura style - situation based) and this is not clarified in the text. Please do so.

5) Effect of gender should be presented, in study 1. 47% were male but only 28 % male in study 2. In general, there is a difference in efficacy beliefs between men and women, men normally score higher irrespective of task.

6) The efficacy-scale is a 4-grade Likert – motivate why not 7-graded as the previous scales?

7) The correlations found between self-efficacy, cheating behaviors, and state boredom are presented as they were causal, on what ground can such a claim be made? Please clarify or rephrase. The directions of these correlations should also be discussed, e.g., test maybe it is the cheating behavior that lower your self-efficacy? The conclusion (line 249) is too definite and needs to be discussed. In general, all conclusions should be discussed, and alternate interpretations should be provided (in the discussion)

8) In figure 1 we get no information about common/shared variance in the three factors, estimate this or at least make a comment.

9) Rephrase this sentence, word order (lines 144-45). Participants  were told that the more gifts they would receive at the end of the experiment, the more 145 correctly identified sentences they solved.

10) The dependent cheating variable only has 3 possible outcomes 0-1-2 where 2 stands for highest cheating value. This variable seems very unsensitive to me – I am really uncertain if this really measures anything of interest, or at least participants’ moral, maybe it measures sustained attention? Other possible confounds could be IQ and gender. This should be discussed as limitations in the discussion. Please create a new paragraph with limitations in the discussion.

11) I would like to see all results from both tests summarized in a table including M, SD, range, p-values and effect sizes (in comparisons).

12) lines 72-73 is a truism, internal traits is caused by internal traits and external states is caused by external conditions, delete or rephrase.

References

Bong, M., & Skaalvik, E. M. (2003). Academic Self-Concept and Self-Efficacy: HowDifferent Are They Really? Educational psychology review, 15(1), 1-40. doi:10.1023/a:1021302408382

Fahlman, S. A., Mercer-Lynn, K. B., Flora, D. B., & Eastwood, J. D. (2011). Development and Validation of the Multidimensional State Boredom Scale. Assessment, 20(1), 68-85. doi:10.1177/1073191111421303

Farmer, R., & Sundberg, N. D. (1986). Boredome Proness - The Development and Correlates of a New Scale. Journal of Personality Assessment, 50(1), 4-17.

Pajares, F., & Schunk, D. (2005). Self-efficacy and self-concept beliefs. New Frontiers for Self-Research, March H. Craven R, McInerney D (eds.). Greenwich, CT: IAP.

Struk, A. A., Carriere, J. S. A., Cheyne, J. A., & Danckert, J. (2015). A Short Boredom Proneness Scale: Development and Psychometric Properties. Assessment, 24(3), 346-359. doi:10.1177/1073191115609996

Author Response

Manuscript: behavsci-1726512

Thank you very much for providing us the opportunity to revise the manuscript. We appreciate the comments and suggestions from the reviewers. For your convenience, the review comments and our replies were appended below. The revisions in the manuscript were also highlighted.

Reviewer 3

Summary and overall comment

This is an empirical paper consisting of two studies. The first study (N=104) investigated if there was a relation between moral and boredom, that is, if bored people have lower moral standards shown as cheating behavior in an unsolvable anagram task. Results revealed that this was the case, bored people cheated more in this task. The second study (N=139) found that “the state” of self-efficacy mediated “the state” of boredom, a high level of boredom leads to lower self-efficacy, which in turn leads to increased levels of cheating behavior. From this it is concluded state boredom predicts cheating behavior that in turn is mediated by self-efficacy.

The questions are of academic interest but introduction and aims needs a major revision. The intro is very short and there are many concepts that are not properly explained, and several relations and causations are suggested that are not empirically underpinned, e.g., state boredom vs. trait boredom, situation based (state) self-efficacy vs. trait self-efficacy. Moreover, if these affect each other independently and of utmost importance, the danger in claiming causation between factors that correlates. The distinction between possible correlations and causations is not properly described. Therefore, I advocate a major revision of the manuscript in accordance with the detailed descriptions below found in major and minor comments.

Major and minor Comments

  • The distinction between state boredom and trait boredom should be mad clearer. The Boredom Proness Scale (Farmer & Sundberg, 1986) should be mentioned and preferably the psychometric properties of this scale  (Struk, Carriere, Cheyne, & Danckert, 2015). If boredom is a rather constant trait or if it can differ considerably between situations this has to be taken to account when conclusions are drawn as in the present study. The reference for study 2 works also fine for this purpose (Fahlman, Mercer-Lynn, Flora, & Eastwood, 2011)

Response:

Thanks for the suggestion! Following your suggestion, we distinguished the state boredom and trait boredom. Please see p.2, where we wrote: State boredom is related with trait boredom but has a distinctive construct from trait boredom (Chan et al., 2018; Neu, 1998; Todman, 2003). Specifically, the trait boredom is an internal psychological characteristic, which focuses on one’ s tendency to become bored. In contrast, state boredom is “the actual experience of boredom in a given moment (Fahlman et al., 2011, p.70)”, which is more influenced by external situational factors (Fahlman et al., 2011). The issue of boredom has been discussed by Neu (1998) in terms of endogenous boredom (boredom that stems from within) and reactive boredom (boredom that stems from the environment). Similarly, Todman (2003) also has distinguished boredom according to situation: situation-independent boredom and situation-dependent boredom. Accordingly, researchers developed the Boredom Proneness Scale (BPS) to measure trait boredom (Farmer & Sundberg, 1989) and Multidimensional State Boredom Scale (MSBS) to measure state boredom (Fahlman et al. 2013), respectively.

  • The distinction between Banduras situation (state) determi (state) determined self-efficacy and the broader trait self-efficacy that is closer to a more general self-concept must be much clearer, see e.g. (Bong & Skaalvik, 2003; Pajares & Schunk, 2005).

Response:

Thanks for the suggestion! Following your suggestion, we distinguished the state boredom and trait boredom. Please see p. 2-3, where we wrote: Bandura (1977, 1986, 1997) identified state self-efficacy as an individual's belief about their capability to accomplish specific performance goals based on an individual's expectations and convictions about what he or she can accomplish under specified circumstances. Thus, research on state self-efficacy emphasizes the contextual role in assessing efficacy (Bong & Skaalvik, 2003). While, the trait self-efficacy refers to the way one perceives himself or herself in particular domains of activity, which is closer to the general self-concept (e.g., Bong & Skaalvik, 2003; Pajares & Schunk, 2005). For instance, when it comes to competence in high-jumping, trait self-efficacy refers to how competent a person is at high-jumping in general (Bong & Skaalvik, 2003). In contrast, state self-efficacy refers to people's belief that they will be able to jump a height successfully under specific conditions, such as, one can high-jump 6 ft when he or she in a jump competition (Bandura, 1986).

  • If you want to link cheating behaviors to one or both above-described concepts there must be room for alternate, third variable, explanations, in particular if you experimentally induce both boredom – through the delivery of boring tasks and cheating behavior through unsolvable tasks. Will these findings be possible to replicate or are they just valid for this particular situation. These clarifications/distinctions should be made clear in the introduction, and alternate explanations should be presented in the discussion.

Response:

Thanks for the suggestion! Following your suggestion, we wrote the alternatives (e.g., cognitive load or ego-depletion) explain cheating behaviors and the limitations of our findings, please see p.8-9 where we wrote: Finally, we primarily aimed to measure the correlation between state boredom and cheating behaviors on task performance, instead of making broad inferences about the correlation by and large. Thus, the present reanalysis mainly illustrates the correlation in the study areas of task performance.

We also made clear our study scope in the introduction, please see p.3, where we wrote: In the current work, we primarily aimed to measure the correlation between state boredom, state self-efficacy and cheating behaviors on task performance because state boredom and state self-efficacy are the actual experience in a given situation or task.

  • Regarding, the self-efficacy scale neither the scale or the reference is not presented in text or in reference list. Here we are talking about a general trait I guess (not Bandura style - situation based) and this is not clarified in the text. Please do so.

Response:

We are sorry that we did not present the reference of self-efficacy scale in our manuscript. We have added these information. Please see reference 46. We also clarified how we adapted the general self-efficacy scale in the updated manuscript, please see p.6, where we wrote: Because we aim to measure state self-efficacy according to the task of cheating behaviors in our study and after the manipulation of state boredom, we adapted the general self-efficacy scale (Schwarzer & Jerusalem, 1995) by revising the instruction of self-efficacy according to the task of cheating behaviors. Specifically, we asked participants to indicate their current belief in their capability to accomplish the anagram task above.

  • Effect of gender should be presented, in study 1. 47% were male but only 28 % male in study 2. In general, there is a difference in efficacy beliefs between men and women, men normally score higher irrespective of task.

Response:

Thanks for the suggestion! Following your suggestion, we conducted a correlation analysis between gender and self-efficacy, results showed that gender was not correlated with self-efficacy. In addition, we included gender as a moderator, and conducted a moderated mediation (state boredom is IV, state self-efficacy is a mediator, and cheating behavior is DV) with model 7 and 14 in process. Results showed that the interaction of gender by state boredom did not predict state self-efficacy in model 7, and the interaction of state self-efficacy by gender did not predict cheating behavior in model 9. Thus, we did not consider gender in our subsequent analyses.

  • The efficacy-scale is a 4-grade Likert – motivate why not 7-graded as the previous scales?

Response:

We used the general self-efficacy scale (Schwarzer & Jerusalem, 1995), in which it is a 4-point scale. In addition, Podsakoff et al. (2003) indicated that the use of the same scale format on a questionnaire can result in Common Method Biases. Thus, we did not revise the 4-point scale to 7-point scale.

7) The correlations found between self-efficacy, cheating behaviors, and state boredom are presented as they were causal, on what ground can such a claim be made? Please clarify or rephrase. The directions of these correlations should also be discussed, e.g., test maybe it is the cheating behavior that lower your self-efficacy? The conclusion (line 249) is too definite and needs to be discussed. In general, all conclusions should be discussed, and alternate interpretations should be provided (in the discussion)

Response:

Sorry for the confusion caused by our description! The correlations between self-efficacy, cheating behaviors, and state boredom were not causal. We rewrote the description, please see p.7, where we wrote: state boredom was indirectly associated with cheating behaviors on task performance through reduced state self-efficacy. In addition, we discussed some alternate interpretation (e.g., ego depletion or cognitive load) in the limitations, please see p.8-9 for details.

8) In figure 1 we get no information about common/shared variance in the three factors, estimate this or at least make a comment.

Response:

Thanks for the suggestion! Following your suggestion, we conducted a Common Method Bias Test and reported the results, please see p.6, where we wrote: A common method bias test was conducted using the Harman’s single-factor test (Podsakoff et al., 2003). The results showed that the KMO value was 0.89 (p < 0.001), indicating that the scales are suitable for factor analysis. There were eight factors with eigenvalues greater than 1, and the first factor explained a variance of 34.10%, which did not reach the critical criterion of 40%. Therefore, the influence of common method bias is not considered to be great in this study.

  • Rephrase this sentence, word order (lines 144-45). Participants  were told that the more gifts

Response:

Thanks for the suggestion! We rewrote the sentence, please see p.4, where we wrote: Participants were told they would receive more gifts at the end of the experiment if they solved more sentences correctly.

10) The dependent cheating variable only has 3 possible outcomes 0-1-2 where 2 stands for highest cheating value. This variable seems very unsensitive to me – I am really uncertain if this really measures anything of interest, or at least participants’ moral, maybe it measures sustained attention? Other possible confounds could be IQ and gender. This should be discussed as limitations in the discussion. Please create a new paragraph with limitations in the discussion.

Response:

Thanks for the suggestion! Following your suggestion, we create a new paragraph in limitations, please see p.8, where we wrote: although the anagram task, which we adapted from Kilduff et al. (2016) and Pierce et al. (2013) commonly used to measure immoral behaviors, the measurement could be confounded by IQ, participants’ interest, attention etc. Further studies could exclude these confound variables in order to verify the present findings.

  • I would like to see all results from both tests summarized in a table including M, SD, range, p-values and effect sizes (in comparisons).

Response:

Thanks for the suggestion! Following your suggestion, we reported M, SD, range, p-values and effect sizes in table 1 and 2.

12) lines 72-73 is a truism, internal traits is caused by internal traits and external states is caused by external conditions, delete or rephrase.

Response:

Thanks for the suggestion! Following your suggestion, we rewrote the sentence,  please see p.2, where we wrote: the trait boredom is an internal psychological characteristics, which focuses on one’ s tendency to become bored. In contrast, state boredom is “the actual experience of boredom in a given moment” (Fahlman et al., 2011, p.70), which is more influenced by external situational factors (Fahlman et al., 2011).

References

Bong, M., & Skaalvik, E. M. (2003). Academic Self-Concept and Self-Efficacy: HowDifferent Are They Really? Educational psychology review, 15(1), 1-40. doi:10.1023/a:1021302408382

Fahlman, S. A., Mercer-Lynn, K. B., Flora, D. B., & Eastwood, J. D. (2011). Development and Validation of the Multidimensional State Boredom Scale. Assessment, 20(1), 68-85. doi:10.1177/1073191111421303

Farmer, R., & Sundberg, N. D. (1986). Boredome Proness - The Development and Correlates of a New Scale. Journal of Personality Assessment, 50(1), 4-17.

Pajares, F., & Schunk, D. (2005). Self-efficacy and self-concept beliefs. New Frontiers for Self-Research, March H. Craven R, McInerney D (eds.). Greenwich, CT: IAP.

Struk, A. A., Carriere, J. S. A., Cheyne, J. A., & Danckert, J. (2015). A Short Boredom Proneness Scale: Development and Psychometric Properties. Assessment, 24(3), 346-359. doi:10.1177/1073191115609996

Round 2

Reviewer 2 Report

I have no further comments

Author Response

Thank you for your time and effort.

Reviewer 3 Report

I thank the authors for this thorough review. I found all my concerns properly addressed and the remaining shortcomings are clearly adressed in the limitation section. 

In all, I endorse publication, just three minor points could still be adressed:

1)    Page 3 I don’t think the high jump example worked very well – I think both were examples of trait, situation based self-efficacy. Please find a better example. The conclusion was ok.

2)    Gender, interesting results that no differences were found, this could be mentioned in the discussion. Could ego-depletion be the cause??

3)    Range – not rang in tables 1 and 2

Author Response

Dear editor,

Thank you very much for providing us with the opportunity to revise the manuscript. We appreciate the comments and suggestions from the reviewer. For your convenience, the review comments and our replies were appended below. The revisions in the manuscript were also highlighted.

I thank the authors for this thorough review. I found all my concerns properly addressed and the remaining shortcomings are clearly addressed in the limitation section. 

In all, I endorse publication, just three minor points could still be addressed:

  • Page 3 I don’t think the high jump example worked very well – I think both were examples of trait, situation based self-efficacy. Please find a better example. The conclusion was ok.

Response:

Thanks for the suggestion! Following your suggestion, we rewrote the example, we hope the example is better than the previous. Please see p.3, where we wrote: For instance, when it comes to academic domains, trait self-efficacy refers to how competent a person perceives about themselves in academic settings, which is fairly stable. In contrast, state self-efficacy refers to a person's belief that he or she is capable of successfully completing academic tasks at a designated level, which is relatively malleable based on the nature of task.

2)    Gender, interesting results that no differences were found, this could be mentioned in the discussion. Could ego-depletion be the cause??

Response:

Thanks for the suggestion! Following your suggestion, we disused the findings, please see p8, where we wrote: Our correlation analyses in Study 2 showed that gender was not correlated with state self-efficacy, which is inconsistent with some previous studies, in which males had significantly higher general self-efficacy than females because males are sometimes overconfident of their abilities and performances when self-recognizing, whereas females do the opposite. However, our findings are consistent with other previous studies. According to Hackett and Campbell (1987), gender-neutral tasks did not reveal any differences between men and women and the possibility is that the nature of the task may influence the self-efficacy expectations. For instance, there is a greater likelihood of gender differences in self-efficacy where there is more gender-role stereotyping in tasks (such as, math examines). Thus, we can presume that inconsistency between our findings and previous finding would also be related to the nature of the task because our cheating task is not relevant to traditional gender-role stereotyping. Consequently, we did not find the correlation between gender and self-efficacy.

3)    Range – not rang in tables 1 and 2

Response:

Many sorry for the typo, we revised it in table 1 and 2.